# Design and Immunological Properties of the Novel Subunit Virus-like Vaccine against SARS-CoV-2

**DOI:** 10.3390/vaccines10010069

**Published:** 2022-01-02

**Authors:** Igor V. Krasilnikov, Aleksandr V. Kudriavtsev, Anna V. Vakhrusheva, Maria E. Frolova, Aleksandr V. Ivanov, Marina A. Stukova, Ekaterina A. Romanovskaya-Romanko, Kirill A. Vasilyev, Nataliya V. Mushenkova, Artur A. Isaev

**Affiliations:** 1Department of Vaccinology, Smorodintsev Research Institute of Influenza of the Ministry of Health of the Russian Federation, 197376 Saint Petersburg, Russia; kiv06@mail.ru (I.V.K.); marina.stukova@influenza.spb.ru (M.A.S.); romromka@yandex.ru (E.A.R.-R.); kirillv5@yandex.ru (K.A.V.); 2Betuvax, 121096 Moscow, Russia; aleks.kudrjavcev@gmail.com (A.V.K.); mushenkova@unicpartners.com (N.V.M.); 3PJSC Human Stem Cells Institute, 129110 Moscow, Russia; maria.frolova92@gmail.com (M.E.F.); art.isaev@genetico.ru or; 4Razvitie BioTechnologiy, 119136 Moscow, Russia; ivanoffal@yandex.ru; 5Center of Genetics and Reproductive Medicine “Genetico”, 119333 Moscow, Russia

**Keywords:** betulin, COVID-19, nanoparticle vaccine, RBD-Fc-based vaccine, SARS-CoV-2

## Abstract

The COVID-19 pandemic is ongoing, and the need for safe and effective vaccines to prevent infection and to control spread of the virus remains urgent. Here, we report the development of a SARS-CoV-2 subunit vaccine candidate (Betuvax-CoV-2) based on RBD and SD1 domains of the spike (S) protein fused to a human IgG1 Fc fragment. The antigen is adsorbed on betulin adjuvant, forming spherical particles with a size of 100–180 nm, mimicking the size of viral particles. Here we confirm the potent immunostimulatory activity of betulin adjuvant, and demonstrate that two immunizations of mice with Betuvax-CoV-2 elicited high titers of RBD-specific antibodies. The candidate vaccine was also effective in stimulating a neutralizing antibody response and T cell immunity. The results indicate that Betuvax-CoV-2 has good potential for further development as an effective vaccine against SARS-CoV-2.

## 1. Introduction

At the end of December 2019, the Chinese authorities announced an outbreak of pneumonia caused by the new Severe Acute Respiratory Syndrome Coronavirus-2 (SARS-CoV-2) in Wuhan. On 11 March 2020, the World Health Organization (WHO) announced a pandemic of a new infection. As of 3 November 2021, the number of cases of coronavirus in the world exceeded 246 million people, and the number of deaths approached 5 million people, reflecting the global health crisis [1]. Previous knowledge of SARS and Middle East Respiratory Syndrome Coronavirus (MERS) vaccines allowed researchers to begin developing a SARS-CoV-2 vaccine just weeks after the outbreak. Although no candidate SARS or MERS vaccine has attained market authorization, this experience has helped to select the target antigen and suitable vaccine platforms.

Like other members of the Coronaviridae family, SARS-CoV-2 is a single-stranded (+) RNA virus whose viral genome is packaged into a helical capsid formed by the nucleocapsid protein (N) and surrounded by an envelope. At least three proteins are known to be associated with the envelope, and spike (S) glycoprotein is a major component [2]. It is a structural protein responsible for the crown-like shape of the viral particles, from which the name “coronavirus” originated.

The S protein is required for receptor binding, membrane fusion, and viral penetration. It is a 140 kDa protein, consisting of 1273 amino acids, similar to the fusion proteins of the class I viral membrane. The S protein is processed at the S1/S2 cleavage site by host cell proteases, thus generating an N-terminal S1-ectodomain and a C-terminal S2-membrane-anchored protein [3]. The S1 domain consists of subdomains NTD (N-terminal domain), RBD (Receptor binding domain; residues 319–527, and S1 subdomains (SD1 and SD2) which are closest to the S2 domain.

RBD is responsible for interaction with angiotensin-converting enzyme 2 (ACE2) [4], the primary receptor for SARS-CoV-2 expressed in many tissues, including type II alveolar epithelial cells in the lungs. The fusion peptide (FP), two heptad repeats (HR1 and HR2), central helix (CH), transmembrane (TM) domain, and cytoplasmic tail (CT) are located in the S2 subunit. The S2 region is involved in fusion between the viral membrane and host cell membranes. The three S1/S2 protomers non-covalently bind to form the functional S-trimer [2].

The spike protein is a known target for host immune defense. An antibody response against the S protein in patients infected with SARS-CoV is observed within 4–8 days after the onset of the first symptoms, while the neutralizing antibody response to the S protein begins to develop by the second or third week [5]. It was shown that RBD is the main target of neutralizing antibodies, and has the highest immunogenicity among the tested recombinant S protein fragments [6]. In addition to the epitopes recognized by antibodies, various immunodominant T cell epitopes have been identified in the RBD sequence [7]. T cells play a critical role in removing and killing the virus-infected cells, and numerous studies have shown their role in the control of SARS and MERS infections [5]. T cell responses have also been found in patients with COVID-19 [7]. The decrease and functional depletion in T cells in patients with COVID-19 is associated with mortality and increased expression of immune-inhibiting factors, such as programmed death-1 (PD-1) and T cell immunoglobulin mucin-3 (TIM-3), which are usually observed in patients progressing from prodromal to clearly symptomatic stages. The activation of T cell responses to SARS-CoV-2 is an important point that must be considered when developing effective vaccination strategies [5].

The surface exposure of S protein, as well as the presence of B and T cell epitopes important for antiviral defense, make this protein a good vaccine candidate. The fragments of S protein used in vaccine design include the full-length S protein, the RBD-domain, the S1 subunit, NTD, and FP [5]. Currently, most of the SARS-CoV-2 subunit vaccines under development are based on RBD [5].

At present 5 vaccines are approved on several markets: Comirnaty (Pfizer/BioNTech, New York NY, USA; Mainz, Germany; Beijing, China), Sputnik V (Gamaleya, Moscow, Russia), Spikevax (Moderna, Cambridge, MA, USA) Vaxzevria (AstraZeneca, Cambridge, UK), BBIBP-CorV (Sinopharm, Beijing, China), CoronaVac (Sinovac, Beijing, China), COVID-19 Vaccine (Janssen, Beaver Dam, WI, USA). In addition to Sputnik V, two other vaccines have received authorization in Russia: KoviVac (Federal Research Center for Research and Development of Immunobiological Preparations) and EpiVacCorona (Vector). Various platforms have been used to develop clinically validated vaccines: Pfizer/BioNTech and Moderna are RNA vaccines expressing COVID-19 spike glycoprotein, while Gamaleya, AstraZeneca, and Janssen express spike protein from adenovirus vectors. Sinopharm and Sinovac are whole inactivated virus vaccines with alum as an adjuvant [8]. According to reported clinical data, the efficacy of current vaccines varies from 79% [9] to 95% [10] thus fulfilling the target product profile for COVID-19 vaccines suggested by WHO, with an estimated 70% endpoint [11]. Despite the existence of approved SARS-CoV-2 vaccines, the activity in the development of new products is still very high. More than 200 candidate vaccines are at various stages of the development, over 50 candidate vaccines were in clinical trials and 18 were at the stage of efficacy testing at the beginning of 2021 [8]. The dependence of mRNA- and adenovirus-based vaccines on the “cold chain” transport makes vaccination in resource-limited settings challenging and creates a need for more stable vaccines. Furthermore, there have been concerns about adverse events following vaccination [12], which also stimulates ongoing vaccine development and improvement. In addition, different platforms and technologies may be suitable in different epidemiological settings and are important for diversification of manufacturing capacity, which is important to ensure sufficient vaccine availability.

We have developed a SARS-CoV-2 RBD-based subunit vaccine (Betuvax-CoV-2) with an improved Fc-linked construction in the form of special virus-like particles (Betuspheres) made of an organic compound. Here we show that Betuvax-CoV-2 is immunogenic in mice, leading to the production of neutralizing antibodies against SARS-CoV-2 and induction of SARS-CoV-2-specific T cell responses.

## 2. Materials and Methods

### 2.1. Preparation of RBD-SD1-Fc Recombinant Fusion Antigen

The antigen plasmid (“Lytic Solutions”, Madison, WI, USA) contains the full-length RBD-SD1 gene sequence with additional flanking amino acids (NCBI Reference Sequence: YP_009724390.1, amino acids 302-640) that fuse with the IgG1 (immunoglobulins of class G1) Fc fragment (NCBI Reference Sequence: 4CDH_A, amino acids 29-254). The RBD-SD1-Fc gene was codon-optimized for expression in CHO cells and synthetically produced by “Atum” (Manchester, NH, USA). The RBD-SD1-Fc gene was cloned between KpnI-XhoI sites in the pcDNA3.1 vector (“Thermo Fisher Scientific”, catalog No. V79020, Waltham, MA, USA) under transnational control of the CMV promoter, and transformed into *E. coli* (SG6328; “Scarab Genomics”, Madison, WI, USA) for plasmid production.

FreeStyle™ CHO-S™ cells (“Invitrogen”, catalog No. R800-07, Waltham, MA, USA) were cultivated in CHOgro™ medium (“Mirus Bio” LLC, catalog No. MIR 6200, Madison, WI, USA), containing up to 8 mM L-glutamine (“Mirus Bio” LLC, catalog No. MIR 6240, Madison, WI, USA) and 0.2% Poloxamer 188 (“Mirus Bio” LLC, catalog No. MIR 6230, Madison, WI, USA). Transfection of cells was carried out with plasmid pcDNA3.1 using the TransIT-Pro system (“Mirus Bio” LLC, Madison, WI, USA), according to manufacturer notes.

The cells were removed from the culture medium by centrifugation at 1500× *g*. The RBD-SD1-Fc protein was affinity-purified by “Lytic Solutions” (Madison, WI, USA) using Protein-A agarose chromatography. Bound protein was washed on resin with phosphate-buffered saline to remove non-specific contaminating protein. The RBD-SD1-Fc was eluted using low pH (100 mM Glycine/HCl pH 2.7) and immediately neutralized with 1/10 volume of 1 M Tris/HCl pH 8. Elution fractions were analyzed by absorbance at 280 nm using a Nanodrop ND-1000 spectrophotometer (“Thermo Fisher Scientific”, Waltham, MA, USA). Peak protein-containing fractions were pooled, pH adjusted to 7.2–7.5 and sterile filtered (0.22 µm PES membrane filter; performed under aseptic conditions in a Class-2 biosafety cabinet) and reanalyzed by UV spectroscopy. A fraction was aseptically taken and used for subsequent analysis, whereas the bulk stored at 4 °C for short-term (weeks) or −20 °C for long-term (months or longer).

The final protein lot was analyzed by SDS-PAGE with Coomassie R-250 staining (“Thermo Fisher Scientific”, Waltham, MA, USA) to determine protein intactness and purity according to the manufacturer’s instructions. Purified RBD-SD1-Fc proteins were evaluated in 4–12% gradient SDS-PAGE under reducing conditions. The wells were filled with 10 μL of the sample. The separation was carried out at a constant voltage of 200 V. For the determination of proteins in the gel, Coomassie G-250 staining was used. The calculation of the purity was performed using the ImageJ program (U. S. National Institutes of Health, Bethesda, MD, USA).

### 2.2. Betulin Nanospheres Production and Characterization

To prepare the adjuvant, we used a betulin solution in tetrahydrofuran (TGF) provided by LLC “Berezovyy Mir” (Moscow, Russia). At the first step, sterilizing filtration was carried out through a nylon membrane (NRG Pall N66+, “Pall Corporation”, New York, NY, USA) with 0.22 µm pore diameter. To obtain a homogenous dispersion of the adjuvant and reduce the concentration of TGF, 1% betulin solution was mixed with a 25-fold volume of sterile 0.05 M Tris-buffer solution (pH 9.0). The suspension was sonicated (35–40 kHz) for 10 min.

The zeta potential of betulin nanoparticles was measured using electrophoretic light scattering (ELS) with Zetasizer Nano (“Malvern Panalytical”, Malvern, Worcestershire, UK). The size of adjuvant nanoparticles was determined using Laser Dynamic Light Scattering (Zetasizer Nano ZS, “Malvern Panalytical”, Malvern, Worcestershire, UK) with a scattering angle of 170° and λ = 633 nm. The calculation of nanoparticle size was performed on the basis of the Stokes–Einstein equation.

The presence of betulin was analyzed using reverse phase HPLC (Shimadzu LC-2010 liquid chromatograph, “Shimadzu”, Kyoto, Japan) and Kromasil 100-C18 (150 × 4.8 mm) analytical column (“Nouryon”, Amsterdam, The Netherlands) with isocratic elution and detection at λ = 210 nm.

### 2.3. Betuvax-CoV-2 Preparation

A recombinant antigen of SARS-CoV-2 RBD-SD1-Fc and a previously prepared adjuvant in 0.05 M Tris-HCl buffer solution with addition of 0.15 M NaCl (pH 7.5) were mixed, so that the final concentration of RBD antigen was 40 μg/mL or 10 μg/mL, and the adjuvant is 400 μg/mL. The components were thoroughly mixed and placed in a shaker in the refrigerator for 3 h. The vials with RBD-SD1-Fc and adjuvant were then stored in a refrigerator at 2–8 °C.

The completeness of RBD-SD1-Fc sorption on betulin particles was assessed using HPLC (Shimadzu LC-2010 liquid chromatograph, “Shimadzu”, Kyoto, Japan) and a Superdex 200 Increase 10/300GL analytical column (“Merck KgaA”, Darmstadt, Germany). Preliminarily, the vaccine samples were centrifuged at 4000× *g* for 30 min.

### 2.4. Transmission Electron Microscopy (TEM)

The betulin spheres were diluted to 1 mg/mL in 0.05 M Tris-HCl (pH 7.4) buffer and sonicated at 42 KHz ± 6% 4 times within 30 s. The samples (3 µL) were applied to copper grids and stained with 2% phosphotungstic acid. For electron microscopy, JEOL JEM 1011 electron microscope (“JEOL” Ltd., Tokyo, Japan) was used.

Vaccine samples containing betulin adjuvant (0.4 mg/mL) in DPBS buffer (pH 7.4) with RBD-SD1-Fc protein (40 μg/mL) were sonicated at 42 KHz ± 6% 4 times within 30 s. Then, 3 μL of the samples were applied to carbon-coated glow-discharged copper grids and negatively stained with 1% uranyl acetate two times for 30 s each. Air-dried grids were imaged on a JEOL 2100 transmission electron microscope (“JEOL” Ltd., Tokyo, Japan) equipped with a Gatan Ultrascan 1000XP CCD camera and operated at 200 kV. The data were collected under low-dose conditions at a ~1.5 μm defocus and 40.000 magnification.

### 2.5. Animal Ethics Statement

The BALB/c and C57BL/6 mice used in this study were derived from the “Stolbovaya” branch of the Federal State Budgetary Institution Scientific center of biomedical technologies of Federal Medical and Biological Agency (FSBI SCBT FMBA of Russia). The number of animals used in the study was sufficient to assess the vaccine immunogenicity, as well as for statistical analysis. At the same time, the number of animals was minimal in terms of ethical principles. The animals were kept under standard conditions in accordance with Directive 2010/63/EU of the European Parliament and of the Council of the European Union, dated 22 September 2010, on the protection of animals used for scientific purposes and in accordance with the sanitary and epidemiological rules SR 2.2.1.3218-14 “Sanitary and Epidemiological Requirements for the Device, Equipment and Maintenance of Experimental Biological Clinics (Vivariums)” (approved by the Resolution of the Chief State Sanitary Physician of the Russian Federation, dated 29 August 2014 No. 51).

### 2.6. Animal Husbandry

Animal husbandry was conducted in accordance with each facility’s SOPs. During the experiment period, the mice were placed in autoclavable polycarbonate cages (“BENEX”, Prague, Czech Republic), type T3A, S = 800 cm^2^, in groups up to 12 individuals, so the area in the cage for one animal was ≥60 cm^2^. Wood pellets Lignocel Wood Fibers (“JRC”, Rosenberg, Germany) were used as bedding. Lighting was provided on a light/dark cycle approximately 12 h each day. The animal room temperature range was 18–22 °C; the relative humidity range was 50–70%. Animals were fed with granular extruded food for mice, rats, hamsters (“Laboratorkorm”, Moscow, Russia), and were given purified water, according to SanPiN 2.1.4.1074-01. Water and chow were available ad libitum throughout the studies.

### 2.7. Animal Study of Humoral Immune Response

BALB/c mice of both sexes (18–20 g, *n* = 16 per group) were immunized by intraperitoneal (ip) injection with two doses of vaccines spaced by 14 days apart (study day 0 and 14) containing 5 μg or 20 μg dose of RBD-SD1-Fc antigen with betulin adjuvant (200 μg) in 0.5 mL of 0.05 M Tris-HCl buffer solution with 0.15 M NaCl (pH 7.5). Two separate groups (*n* = 10) received one and two immunizations with 5 or 20 μg of RBD-SD1-Fc without adjuvant. The placebo group received 0.9% NaCl and served as a non-immunized control (Table 1). Blood samples were collected on the 14th day after a single immunization and on the 14th day after a double immunization. Blood samples were obtained from the mandibular vein and stored at room temperature for 2 h, and then they were centrifuged for 10 min at 5000× *g*. The resulting whole serum was collected. Until the moment of use, the serum was stored at a temperature of −20 °C. 

### 2.8. Anti-SARS-CoV-2 S IgG by ELISA

The presence of specific IgG was determined in serum samples of mice after single and double immunizations. Indirect solid phase ELISA (enzyme-linked immunosorbent assay) was used to analyze the presence of IgG antibodies. The enzyme immunoassay was performed on 96-well microtiter plates (“Greiner Bio-One”, Kremsmünster, Austria). The SARS-CoV-2 RBD antigen (1 μg/mL) was applied to the microplate and incubated for 24 h. Then the plates were washed twice with a solution of 0.1% TWEEN20 in PBS. After washing, the plates were incubated with a blocking buffer PBS containing 5% powdered milk (“TF Ditol”, Saint Petersburg, Russia) for 2 h at room temperature and washed again. Mouse serum samples were serially diluted and added to blocked plates prior to incubation at room temperature for 1 h. Following the incubation, plates were washed with PBS three times. Goat anti-mouse IgG (H + L) Strong Zyme HRP (Horseradish Peroxidase) labeled secondary antibodies (“SDT”, Baesweiler, Germany) were added for 1 hour. After incubation with secondary antibodies, the plates were washed four times.

A solution of TMB (3,3,5,5-tetramethylbenzidine) (“Vekton”, Saint Petersburg, Russia) was used as a colored substrate for 10 min at room temperature, then the reaction was stopped by adding 5% sulphuric acid (“Vekton”, Saint Petersburg, Russia). The results of the enzyme immunoassay were recorded on a PR 2100 spectrophotometer (“Sanofi Diagnostics Pasteur”, Lyon, France) at 450 nm.

### 2.9. SARS-CoV-2 Neutralization Assay

BALB/c mice (18–20 g, *n* = 5 per group) were immunized twice by ip injection with RBD-SD1-Fc antigen with betulin adjuvant (200 μg) in 0.5 mL of 0.05 M Tris-HCl buffer solution with 0.15 M NaCl (pH 7.5). Blood samples were collected 14 days after the first and the second immunizations. The placebo group (*n* = 3) received PBS and served as a non-immunized control.

For neutralization, serum samples were diluted with sterile DPBS in 1:4 ratio and incubated for 30 min at 56 °C. After that, serum dilutions were prepared in MEM + 2% FBS in a volume of 50 µL, starting with 1:10. An equivalent volume of SARS-CoV-2 isolate containing 25 TCID50 (median tissue culture infective dose) was added, and the mixture was incubated for 1 h at 37 °C. Dilutions were transferred to a 96-well V-bottomed plate seeded with 1.5 × 10^5^ Vero E6 cells per well in 10% FCS/MEM. After 4 days of incubation at 37 °C cytopathic activity was analyzed. Neutralizing titers were defined as the highest serum dilution at which no cytopathic activity was observed.

### 2.10. SARS-CoV-2 T Cell Response Assay

To assess the T cell response, groups of C57/black SPF mice (*n* = 5 per group) were immunized by ip injection with two doses of Betuvax-CoV-2 containing 5 μg or 20 μg of antigen on days 0 and 22. Spleens were collected 28 days after the second immunization. A non-vaccinated group (*n* = 3) served as a control. Spleen homogenates were filtrated (70 µm pores), washed with DPBS containing 2.5% FCS (Gibco, “Thermo Fisher Scientific”, Waltham, MA, USA), deprived of red blood cells by RBC Lysis Buffer (“BioLegend”, San Diego, CA, USA) and washed with DPBS (2.5% FCS). Splenocytes were seeded on flat-bottomed 96-well plates at a density of 2 × 10^6^ cells, and were either stimulated or not with SARS-CoV-2 S-protein peptide pool (PepTivator^®^ SARS-CoV-2 Prot S, “Miltenyi Biotec”, Bergisch Gladbach, Germany) for 6 h in the presence of anti-CD28 (“BioLegend”, San Diego, CA, USA) and Brefeldin A (“BD Biosciences”, Heidelberg, Germany). Cell phenotype was studied with CD8-PE/Cy7, CD4-PerCP/Cy5.5, CD44-BV510, CD62L-APC/Cy7, IFNγ-BV780, TNFα-BV421, IL2-PE (“BioLegend”, San Diego, CA, USA). Dead cells were detected with Zombie Red (BioLegend”, San Diego, CA, USA). True Stain, containing anti-CD16/CD32 (“BioLegend”, San Diego, CA, USA), was used for blocking non-specific binding. Cell staining was performed with Cytofix/Cytoperm (“BD Biosciences”, Heidelberg, Germany) according to the instructions.

### 2.11. Statistical Analysis

Data were analyzed with Microsoft Office Excel 2010 (“Microsoft”, Redmond, WA, USA) and GraphPad Prism v6.01 software (GraphPad Software, San Diego, CA, USA). Geometric mean, standard deviation, arithmetic mean and standard errors of the mean were calculated. No samples or animals were excluded from the analysis. Randomization was not performed for the animal studies. Samples and animals were not blinded before performing each experiment. The Kruskal–Wallis H test, or its parametric equivalent one-way analysis of variance (ANOVA), was used for analysis of variance between group means, with subsequent pairwise comparison using Tukey test or non-parametric pairwise multiple comparisons by Dunn test. Values *p* < 0.05 were accepted as statistically significant.

## 3. Results

### 3.1. Generation of Recombinant Nanoparticle Vaccine Betuvax-CoV-2

The RBD domain of the SARS-CoV-2 spike protein (Figure 1A) is a validated component of SARS-CoV-2 vaccines [5] and the main target for the development of effective COVID-19 antibodies. Being the critical sequence for interaction with human cells, it contains both epitopes for antibody neutralization [13] and T cell recognition [7].

The structure of Betuvax-CoV-2 antigen includes two domains of the SARS-CoV-2 spike protein: RBD (S319-S527) and SD1 (S528-S590) with linker sequences (Figure 1B). Subdomain SD1 takes part in the formation of two main conformations of prefusion spike protein—the so-called “up” and “down” states. The “up” state is believed to allow binding of the virus to ACE-2 receptors on human epithelial cells, whereas the “down” state is thought to be inactive [14].

Thus, the presence of both RBD and SD1 sequences ensures the assembly of the RBD domain in its native conformation, which is important for virus–host interaction and accordingly for the induction of broader neutralizing responses specific for both conformations.

The Fc-fragment of IgG1 immunoglobulin was fused with the antigenic part of RBD with native flanking regions (Figure 1C) to provide more efficient delivery of the antigen-presenting cells (APCs) through FcR-dependent endocytosis.

The results of SDS-PAGE analysis of the antigen showed two components with a molecular weight of 84 ± 10 kDa and 25 ± 10 kDa (Figure 1D). The main 84 kDa product corresponds to the expected molecular weight of RBD-SD1-Fc protein, with the minor component resembling the products of its degradation. According to SDS-PAGE data, the purity of the pharmaceutical substance is no less than 97%.

An adjuvant based on betulin was used to enhance the antigenic properties of the vaccine. Betulin is a pentacyclic triterpenoid alcohol (Figure 2A) naturally occurring in many plants, especially in birch bark [16]. A method for betulin-based adjuvant production was described in a patent RU2749193. In Betuvax-CoV-2, betulin is present in the form of nanoparticles with a size about 100–180 nm (Figure 2B,C) and has zeta potential −44.3 mV. This negative potential may enhance interaction of nanoparticle with antigen. Size as well as zeta potential are known to be the main characteristics of nanoparticular adjuvants. It has been shown that particles less than 500 nm are effectively phagocytosed by macrophages [17], which leads to increased antigen presentation and immune response.

The RBD-SD1-Fc antigen molecules are adsorbed on the surface of betulin nanoparticles, thereby mimicking the SARS-CoV-2 virus (Figure 3A–C) with similarly sized parameters [18]. The completeness of sorption of RBD-SD1-Fc antigen was assessed using HPLC (Figure 3D,E). RBD-SD1-Fc protein in the supernatant was monitored by the presence of a peak with the retention time of the main component close to the retention time of the reference sample (RBD-SD1-Fc substance). The absence of a peak with a retention time of 14.8 min indicates the completeness of the antigen sorption on the adjuvant particles.

### 3.2. Humoral Immune Response in Mice following Betuvax-CoV-2 Vaccination

Our group has previously shown that the adjuvant properties of betulin are significantly superior to that of the standard aluminum oxide adjuvant (Al(OH)_3_) for B cell immune response (unpublished data). The immunogenicity of RBD-SD1-Fc antigen with or without betulin adjuvant was assessed in mice. BALB/c mice of both sexes were immunized with 5 or 20 µg of antigen and 200 µg of adjuvant. The intraperitoneal (ip) immunization was performed twice with an interval of 14 days. The presence of IgG antibodies to SARS-CoV-2 RBD was analyzed 14 days after the first and the second immunizations. Betuvax-CoV-2 as well as non-adjuvanted antigen elicited elevated RBD-specific IgG antibody responses, reaching the highest levels after the 2nd vaccination (Figure 4). Comparison of the antibody response with and without a betulin adjuvant showed that the addition of betulin resulted in higher and more stable antibody titers in comparison to the non-adjuvanted vaccine. The geometric mean of the antibody titer (GMT) in Betuvax-CoV-2 groups reached 1:20,700 at the highest dose, while in non-adjuvanted vaccine groups the GMT was 1:11,000 or less. A significant difference from the negative control was achieved in the group with the highest dose of Betuvax-CoV-2 after the first immunization, and both doses after the second one.

The Betuvax-CoV-2 efficacy in terms of virus neutralization response was analyzed in mice. BALB/c mice were injected ip with 5 µg of Betuvax-CoV-2, and the presence of neutralizing antibodies (nAbs) was assessed 14 days after the first and the second immunizations. The titers of SARS-CoV-2-nAbs did not change after the single immunization, but significantly increased after the second vaccination (Figure 5). Although the minimal dose of antigen was used, a neutralizing response was induced in four out of five mice with maximal titers of 1:45 and GMT 1:16.

### 3.3. Betuvax-CoV-2 Induces CD4+ and CD8+ Tem Immune Responses in Mice

T cells play a critical role in the fight against SARS-CoV-2 infection and in the formation of immunological memory following recovery from COVID-19. Circulating SARS-CoV-2-specific CD8+ and CD4+ T cells were identified in ~70% and 100% of patients with COVID-19 and spike-specific responses positively correlated with anti-SARS-CoV-2 IgG titers [20]. As was published recently, 93% and 97% of CD4+ and CD8+ T cell epitopes are 100% conserved across SARS-CoV-2 variants, so T cell reactivity induced by vaccination may provide broad protection [21]. Evaluation of natural T cell responses to SARS-CoV-2 in COVID-19-recovered patients has led to the identification of two immunodominant epitopes in the RBD sequence. They corresponded to S316-331 and S320-335 fragments [7]. Both of these sequences are present in Betuvax-CoV-2 antigen, allowing the T cell recognition.

Size as well as the shape of the antigen-carrying particles may direct the type of T helper (Th) cell polarization. Thus, a round shape and size within the range of 100–200 nm was shown to induce Th1-biased response, whereas rod-shaped particles with 1500 nm size stimulated Th2-biased polarization [22]. Therefore, according to the size and shape of vaccine particles, we can expect Th1-polarization of vaccine-induced T cell response.

In mice, T cells can be divided into naïve and memory cells, based on expression of adhesion molecules CD44 and CD62L. Central and effector memory cells have been identified among the memory T cell population. Central memory T cells (Tcm) are critical for the control of systemic viral infections, and they respond more vigorously to secondary challenges than T effector memory (Tem) cells in terms of expansion and virus clearance. However, in the case of lung infections, Tem cells may be more important, as recall responses are equal to, or greater than, those mounted by central memory T cells, and Tem are more efficient in generating second generation of memory T cells [23].

To study if our immunization regimens generated Tem cells C57BL/6 mice were immunized twice with Betuvax-CoV-2 (5 µg or 20 µg ip) with 21-day interval. Splenocytes were isolated 28 days after second immunization, incubated with a SARS-CoV-2 spike peptide pool for 6 h and then analyzed using flow cytometry. The relative frequency of CD8+ and CD4+ T lymphocytes with phenotype of effector memory cells (CD44+ CD62L−) was determined. A significant increase in spike-specific CD4+ and CD8+ Tem cells was measured after two immunizations with the 20 µg dose, while vaccination with 5 µg led to a statistically significant increase in only CD4+ Tem. There was a tendency for dose-dependent growth of Th1-cytokine (IFNγ, IL-2, TNFα)-producing cells among both populations of Tem. The effect reached significance for IL-2-producing Tem (IFNγ− IL-2+ TNFα−) and for multifunctional CD8+ T cells (IFNγ+ IL-2+ TNFα+) in both dose groups (Figure 6). In the high dose group, the growth of IL-2 production by CD8+Tem was also significant.

## 4. Discussion

The advantages of subunit vaccines include ease of manufacture, low cost and low safety risks, as they do not contain genetic material and have a low probability of inducing severe adverse reactions. SARS-CoV subunit vaccines were reported to be safer than other vaccine types, such as those using virus-like particles (VLP) from virus proteins, inactivated whole viruses and rDNA-expressed S-protein, which have been shown to induce hypersensitivity to SARS-CoV components in mouse models of SARS, resulting in Th2-type immunopathology [24]. Several vaccines expressing the full-length SARS-CoV-2 S protein have been approved worldwide [8]. However, the full-length spike protein immunogen contains many non-neutralizing epitopes, and carries a high risk of antibody-dependent enhancement (ADE) effect, which may occur due to the FcγR-mediated internalization of antibody-coated virions in FcγR-expressing cells [25]. The ADE was observed for SARS-CoV, MERS-CoV, HIV-1, Zika and dengue virus vaccinations, and has also become a concern in SARS-CoV-2 vaccine development [24]. However, the risk of ADE is lower for RBD-based vaccines. There was no evidence of ADE activity in anti-RBD sera from mice immunized with recombinant RBDs [26,27].

Furthermore, the full-length SARS-CoV-2 spike protein was shown in an in vitro V(D)J-reporter assay to significantly impede V(D)J recombination, important for B cell response [28]. Although these data have not been proved in vivo, they are in good accordance with delayed and weak adaptive immune responses in patients with severe COVID–19, as well as with data on fewer antibody titers induced by full-length spike-based vaccines compared to the RBD–based vaccines [28].

Certain safety risks are associated with S1 subunit, as it was shown to interact with platelets and fibrin(ogen) directly to cause blood hypercoagulation and impairment of fibrinolysis [29]. S1 presence in circulation may potentially lead to the formation of persistent microclots and blood flow disturbance, thus leading to concerns regarding the safety of S1-based approaches.

Currently, most of the SARS-CoV-2 subunit vaccines under development are based on RBD [5]. This domain has been also widely used in the development of candidate vaccines against SARS-CoV and MERS-CoV [5]. Based on its high homology to SARS-CoV, it is confirmed that SARS-CoV-2 RBD contains immunodominant epitopes capable of producing antibodies that can neutralize viral infection and block viral entry through competitive hACE2 binding. The presence of T cell epitopes within SARS-CoV-2 RBD was identified by computer prediction and confirmed experimentally [7]. RBD-based SARS-CoV vaccines containing S318–S510 were shown to induce both neutralizing antibody responses as well as CD8+ T cell responses, important for viral control [30]. The activation of T cell responses has also been described for experimental SARS-CoV-2 RBD vaccines [27]. Compared to full-length S, the SARS-CoV-2 RBD immunogen elicited a higher titer of neutralizing antibodies with 5-fold higher affinity [31].

The human IgG Fc fragment in the RBD-based vaccines, RBD-Fc, can act as an important immunopotentiator to enhance the immunogenicity of RBD, and the fusion of RBD or its fragments with an Fc region is also a known strategy in SARS-CoV and MERS-CoV vaccine construction [32,33]. RBD-Fc fusion was shown to elicit a higher neutralizing antibody titer than RBD itself [24]. Five different versions of truncated RBD fused to the human IgG Fc fragment have been validated as effective vaccine candidates, among them are S350-588-Fc, S358-588-Fc, S367-588-Fc, S367-606-Fc, and S377-588-Fc [34,35,36,37,38,39].

In this study, the full-length RBD-SD1 with flanking sequences (S302-640) was fused to human IgG Fc for antigen construction. The RBD-SD1 antigen contains five potential B-cell epitopes, as well as eight CD4+ T and ten CD8+ T epitopes [40]. The inclusion of SD1 is important for flexibility of RBD and its natural switching between two physiological states, which allows the production of antibodies for both RBD conformations. Additionally, it has been suggested that neutralizing epitopes in SARS-CoV-2 RBD are conformational rather than linear, indicating the importance of maintaining the correct domain conformation [40]. In the present study, we did assess the distinct contribution of antibodies specific for RBD conformational states to the virus neutralization reaction, but based on the published data on the role of SD1 [14], we can expect an additive role of both domains for effective SARS-CoV-2 neutralization.

To increase the immunogenicity of the vaccine, betulin was used as an adjuvant. Betulin was first isolated in 1788 and used for centuries in traditional medicine. Many studies have shown the antiviral, antifungal and, above all, anticarcinogenic activity of betulin, as well as its hydrophilic derivative, betulinic acid [14,16]. Although data on the immunomodulatory function of betulin are far from complete, it has been clearly shown that betulin has pleiotropic immunostimulatory activity. It stimulates the proliferation of human peripheral blood lymphocytes [41]. In bacterial lipopolysaccharide (LPS)-activated murine dendritic cells, betulin increased the secretion of IL-12p70 [42], and nanoformulation of betulin lead to a further significant increase in LPS-induced IL-12 secretion [16]. Immunomodulatory effects of betulin have also been described in macrophages, although the data is inconclusive, as, depending on the model, betulin possessed pro-inflammatory (increased secretion of IL-6, MCP-1, COX-2) as well as anti-inflammatory activities (decreased levels IL-6 and TNFα, up-regulated IL-10) [16].

Although numerous properties of betulin have been revealed in vitro and in vivo, its clinical application is severely limited due to its insolubility in water, so all pharmacological experiments and applications must be conducted by dissolving triterpene in organic solvents. Previously, several attempts to increase the solubility of betulin were described, including its conjugation with carboxylic acid chains on carbon nanoparticles (f-CNT-Bet) [43] and oligo saccharide complexation [44,45]. In recent years, significant efforts have been made to isolate or synthesize hydrolyzed, fluorinated, nitrated and acetylated betulin derivatives. A nanosystem-entrapped formula of betulin was also studied [46].

The macrophage-stimulating activity of self-assembled nano-sized betulinic acid has been described as leading to an increase in the priming of CD8+ and CD4+ T cells and an IgG antibody response [47]. Although the data on immunomodulatory activity of betulin derivatives are scarce, it has been shown that they may lack some immunostimulatory mechanisms [16]. Currently, studies concerning the adjuvant role of nano-sized betulin or its derivatives are also limited.

We developed the proprietary method of betulin formulation as nanoparticles, with the size mimicking the size of the virion (Betusphere), and this clearly showed that Betusphere addition potentiates stable anti-RBD responses, resulting in an almost two-fold increase in IgG antibody titers compared to non-adjuvanted vaccine. The betulin adjuvanted vaccine was effective in inducing the SARS-CoV-2 neutralization response after two immunizations with a minimal dose. Taking into account the antigen dose and the number of immunizations achieved, levels of neutralizing antibodies are comparable to other RBD-based vaccines [28]. Based on published data [28,48], an increase in nAbs can be expected with increasing vaccine doses and the number of immunizations.

Memory T cells (CD4+ and CD8+) are known to be critical for virus clearance, with CD8+ Tem cells being the most relevant in cases of lung infections [23]. Here we examined the induction of Tem cells in response to Betuvax-CoV-2. We observed an increase in both CD4+ and CD8+ Tem spike-specific cells after two immunizations with an antigen dose of 20 µg. Based on the type of antigen formulation (betulin adjuvant, size and shape of nanoparticles), we expected to find Th1-directed polarization of the cytokine response; indeed, such a trend was observed, and a multiple increase in the amount of CD4+ Tem with IL-2 and TNFα staining was determined. A statistically significant increase was also shown for multifunctional CD8+ T cells (IFNγ+ IL-2+ TNFα+). Our T cell response data are in line with published data for other RBD-based vaccines. For example, in the study of the RBD S319–545 vaccine, a several-fold increase in CD4+ CD44^high^ IFNγ+ and CD8+ CD44^high^ IFNγ+ cells was observed [27]. Clinical data also revealed that SARS-CoV-2-reactive T cells were overwhelmingly Th1, often with a multifunctional IFNγ, IL-2 and TNFα profile [49]. This cytokine pattern is thought to be beneficial, because patients with severe COVID-19 are usually characterized by decreased IFNγ levels and a shift to a more Th2 profile [50].

## 5. Conclusions

The design of a novel SARS-CoV-2 subunit vaccine based on the use of a full-length RBD domain together with SD1 and flanking sequences is here described. The antigen is fused with the Fc and is adsorbed on the surface of virus-size betulin nanoparticles, increasing its immunogenicity. Betuvax-CoV-2 showed improved immunogenicity in mice compared to non-adjuvanted antigen, with efficient induction of humoral and CD4+ and CD8+ Tem responses after two vaccinations. Further studies are needed to evaluate the protective capacity of the vaccine using in vivo models of SARS-CoV-2 infection, and to study its efficacy against the novel variants of concern, including B.1.1.529, named Omicron. We assume Betuvax-CoV-2 has good potential to be developed as an effective preventive vaccine against SARS-CoV-2 and that it can be easily adapted to new coronavirus strains.

## Figures and Tables

**Figure 1 vaccines-10-00069-f001:**
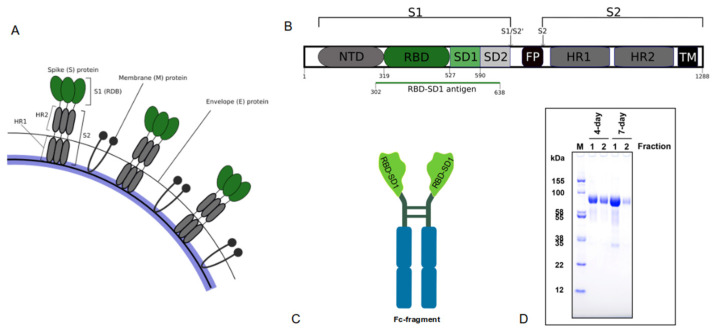
RBD-SD1-Fc antigen construction. (**A**) Schematic structure of SARS-CoV-2 envelope; (**B**) Graphical representation of the human S protein with signal peptide (SP), N-terminal domain (NTD), receptor-binding domain (RBD), SD1 and SD2 subdomains, fusion peptide (FP), heptad repeat 1/2 (HR1/2), and the transmembrane domain (TM). The cleavage sites are indicated by arrows. Adapted from [15]. Sequence cloned for antigen construction is indicated by green line; (**C**) Construction of Betuvax-CoV-2 antigen; (**D**) SDS-PAGE analysis of purified RBD-SD1-Fc protein (M—marker), 10 µg load.

**Figure 2 vaccines-10-00069-f002:**
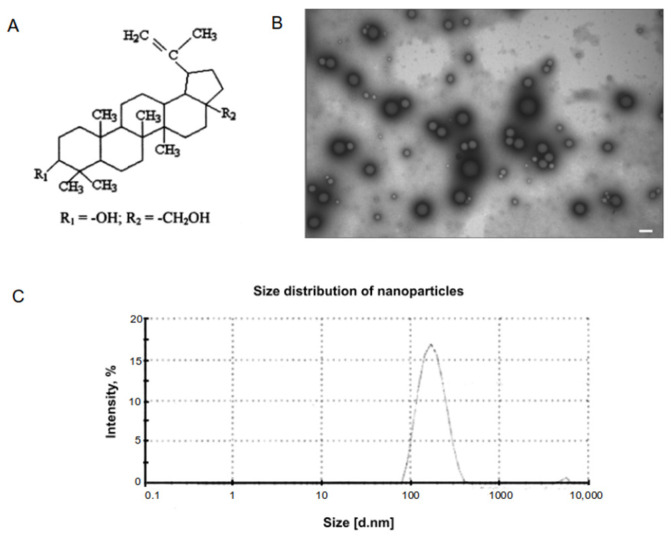
Characterization of betulin-based nanoparticle adjuvant. (**A**) Chemical structure of betulin; (**B**) TEM of betulin nanoparticles. Bar corresponds to 200 nm; (**C**) Betulin particle size analysis by Laser Dynamic Light Scattering.

**Figure 3 vaccines-10-00069-f003:**
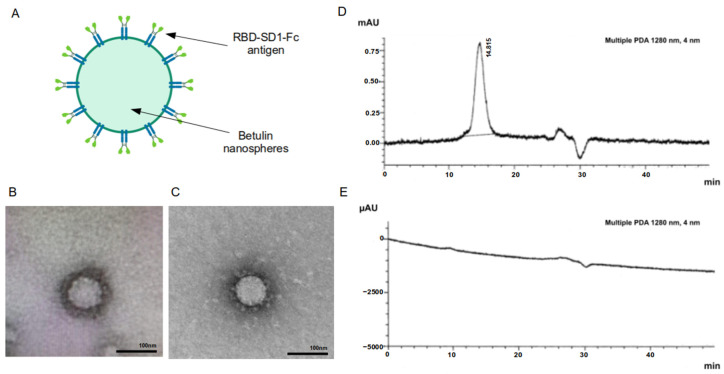
Formation of antigen-coated Betuvax-CoV-2 nanoparticles. (**A**) Graphic representation of Betuvax-CoV-2 nanoparticle; (**B**) Transmission electron microscopy of the Betuvax-CoV-2 showing betulin particle covered with adsorbed antigen, the bar size in 100 nm; (**C**) Typical morphology of SARS-CoV-2, the figure was adapted from [19]; (**D**) Chromatographic profile of a sample of RBD-SD1-Fc antigen preparation and (**E**) a test vaccine sample (supernatant). A peak in the vaccine sample with a retention time of 14.8 min corresponds to unbound antigen, the absence of the peak in the supernatant indicates complete antigen adsorption.

**Figure 4 vaccines-10-00069-f004:**
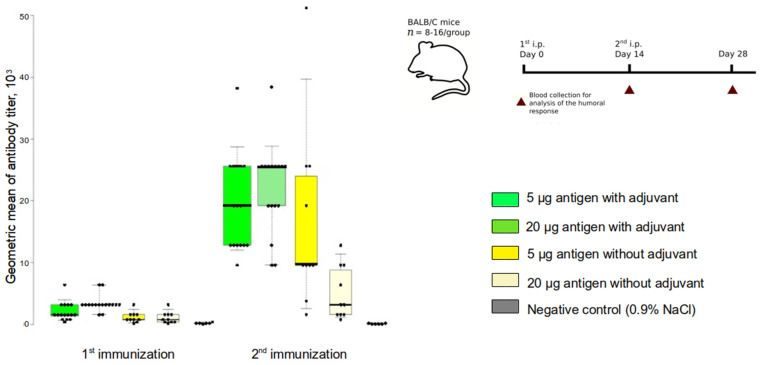
Induction of RBD-specific IgG response following vaccination. The experimental scheme: BALB/c mice of both sexes were vaccinated at Day 0 and Day 14. The presence of anti-SARS-CoV-2 IgG antibodies was analyzed on Day 14 and Day 28. A box and whisker plot of anti-SARS-CoV-2 RBD IgG titers in the serum of immunized mice was made in BoxPlotR; boxes indicate geometric mean titer (GMT) of anti-SARS-CoV-2 RBD-specific IgG titers; Altman whiskers extend to the 5th and 95th percentiles; points indicate individual values; median is shown as horizontal line.

**Figure 5 vaccines-10-00069-f005:**
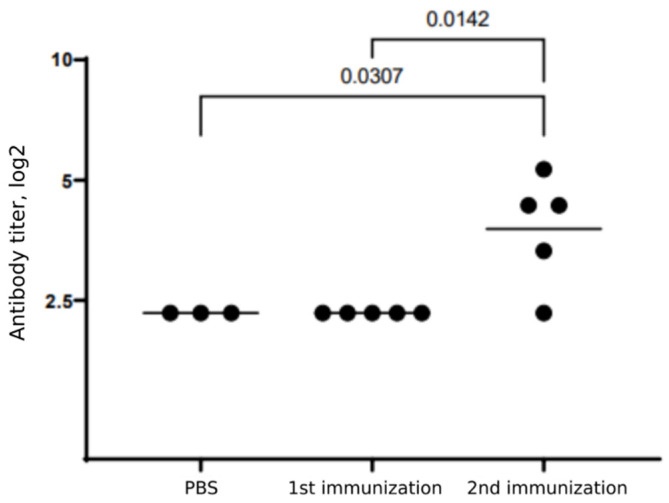
Neutralizing antibody responses in mice following single or double immunization with Betuvax-CoV-2. BALB/c mice were injected ip with 5 µg Betuvax-CoV-2. Points indicate the individual values, horizontal line—GMT for each group. Statistical analysis was performed using ANOVA with subsequent pairwise comparison using Tukey test.

**Figure 6 vaccines-10-00069-f006:**
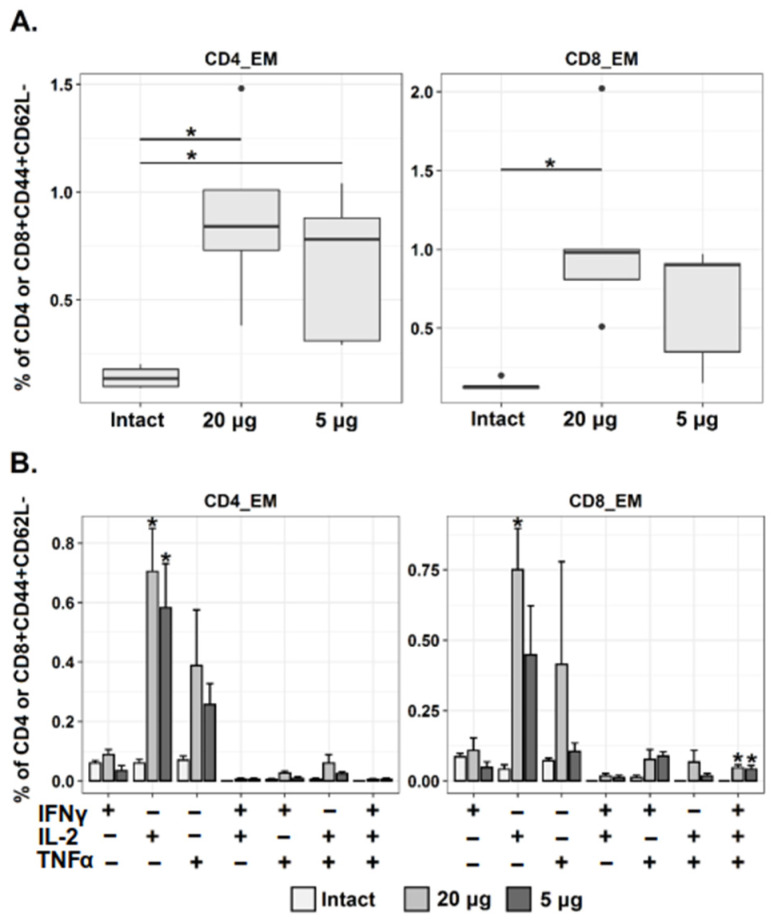
Analysis of SARS-CoV-2-specific CD4+ and CD8+ Tem responses following Betuvax-CoV-2 vaccination. Mice were immunized ip twice with Betuvax-CoV-2 (5 and 20 µg of antigen). (**A**) Percentage of virus-specific CD4+ and CD8+ Tem among CD4+/CD8+CD44+CD62L− splenocytes; points indicate individual outlier values; (**B**) Relative frequency of virus-specific cytokine-producing CD4+ and CD8+ Tem. The bar-plots represent the average percentage of each cytokine-producing population (IFNγ+ IL-2− TNFα−, IFNγ− IL-2+ TNFα−, IFNγ− IL-2− TNFα+, IFNγ+ IL-2+ TNFα−, etc.) within the total CD4 or CD8 Tem cell subset (Mean ± SE) after 6 h of stimulation with SARS-CoV-2 spike protein peptide pool; * *p* < 0.05.

**Table 1 vaccines-10-00069-t001:** The description of animal groups in the humoral immune response study.

Group No.	Intervention	N of Animals
1	Betuvax-CoV-2: RBD-SD1-Fc antigen 5 μg + betulin 200 μg	16
2	Betuvax-CoV-2: RBD-SD1-Fc antigen 20 μg + betulin 200 μg	16
3	RBD-SD1-Fc antigen 5 μg	10
4	RBD-SD1-Fc antigen 20 μg	10
5	Control, 0.9% NaCl	8

## Data Availability

The datasets generated or analyzed during this study are available from the corresponding author on reasonable request.

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
