# Peer review of "Design and Immunological Properties of the Novel Subunit Virus-like Vaccine against SARS-CoV-2"

_vaccines, 2022, doi:10.3390/vaccines10010069_

Round 1
Reviewer 1 Report
The paper is generally well-written and interesting. As the paper makes clear, many groups world-wide are engaged in very similar research to generate useful anti-SARS-CoV-2 vaccines. It is very likely that additional vaccines will be important for many viruses and virus variants.
The data and figures are informative and reasonable.
I thought in the Conclusions the authors might mention that a vaccine against SARS-CoV-2 variants, such as omicron, could easily be designed based on their approach.
Betuvax-CoV-2 was tested in a research vaccine trial in mice. The RBD (receptor-binding domain) and SD1 domains of the spike (S) protein were fused to a human IgG1 Fc fragment and packaged on the surface of betulin adjuvant spherical particles with a size of 100–180 nm (the approximate size of SARS-CoV-2). The size of the nanoparticles was selected to encourage a T-cell response, which was observed. The SARS-CoV-2 human cell receptor is ACE2 (angiotensin-converting enzyme 2) to which RBD binds during infection. Betulin is a natural product initially extracted from birch trees and may have some advantages over other more commonly used adjuvants.
Are any of the paper authors associated with Patent RU2749193 describing the Betulin nanosphere particles? If so, this conflict of interest must be disclosed. Some authors do not understand conflict of interest rules.
Minor points:
Line 518: pattern is thought to be beneficial, since because patients with severe COVID-19 patients are usually characterized by decreased IFNγ level and a shift to a more Th2 profile [50]. “because” is better than “since”
Line 16: Abstract: The COVID-19 pandemic is still ongoing and the need for safe and effective vaccines to prevent infection and control spread of the virus remains urgent. “still ongoing” is redundant
Line 33: number of deaths approached to 5 million people, which is reminiscent (choose another word, i.e., reflects the scale?) of the (immense) global health crisis
Line 415: reported to be safer than other vaccines types such as virus-like particles (VLP)
Author Response
Dear Reviewer,
Thank you for evaluation of our paper and useful comments.
According to your advice we have modified the phrase in line 526 of the Conclusion: Further studies are needed to evaluate protective capacity of the vaccine using in vivo models of SARS-CoV-2 infection and to study its efficacy against the novel variants of concern, including B.1.1.529, named Omicron.
We are sorry, we have not understood conflict of interest rules. This information was correctly added (lines 552-558)
We have also made corrections in lines 16, 33, 415, 518
Reviewer 2 Report
In this study, the authors reported the the development of a subunit vaccine (Betuvax-CoV-2) against CVOID19, and this vaccine is based on RBD and SD1 domains of the spike protein fused to a human IgG1 Fc fragment. The authors showed that immunizations of Balb/c mice (2 doses) with the subunit vaccine induced high titers of RBD-specific antibodies, elicited neutralizing antibody response and T-cell responses, suggesting that this vaccine could be potential effective candidate against SARS-CoV-2.
Although the title is promising, I have some concerns especially about the quality of figures
1- Figure 4 " Induction of RBD-specific IgG response following vaccination", I would suggest the authors to present the Geometric mean of antibody as dot plot not column. Also what error bar means in this figure.
I have a concern about high variation in " 5ug antigen without adjuvant" in the second immunization. Also why 5ug antigen without adjuvant showed more Ab induction than 20ug antigen without adjuvant in the second immunization?
Also statistic is not clear. Is there any significant difference between Betuvax-COV-20ug antigen and Betuvax-COV-5ug antigen ?
2-Figure 5: "Neutralizing antibody responses in mice following single or double immunization with Betuvax-CoV-2". This figure is not convincing. First I would suggest the authors to include the neutralization assay curve using antibody from these mice. Second , the authors need to revise the statistic of this figure, since there is not difference between PBS and 1st immunization, while the p values are varied when compared 2nd immunization vs 1st immunization & 2nd immunization vs PBS.
3-Figure 6; "Analysis of SARS-CoV-2-specific CD4+ and CD8+ Тem responses following Betuvax-CoV- 2 vaccination". Figure 6B is not clear, I do not understand it. Also, Did the authors measure the following cytokines IFNg, IL-2, and TNF-a in the plasma of these mice, or at least the transcript level of these cytokines in the spleen?
Author Response
Dear Reviewer,
Thank you for these helpful comments, we took them into account and improved our article. Please, see our answers in the attachment.

Reviewer 3 Report
The paper reports a vaccine design basing on the fusion of the receptor binding domain (RBD) and the SD1 region of the spike protein of SARS-CoV-2 with the Fc fragment of IgG1. The authors showed that the construct could trigger the production of neutralization antibodies in mice (although the error bars shown in Figure 4 were large). In addition, they showed that adsorbing the construct to virus-size betulin nanoparticles further improved immunogenicity. They also examined the ability of their constructs to induce CD4+ and CD8+ T cell responses, and the results were positive. The presentation was clear, although some parts of the Discussions could have gone into the Introduction. Although increasing the sample size could provide clearer support for their conclusions, it is probably acceptable in its present form as it introduced constructs that might be refined into practically useful vaccines.
Author Response
Dear reviewer,
Thank for evaluation of our paper and your comments. We agree that it is important to use larger groups of animals to achieve more precise results. In fact we hope that the effect of immunization on IgG response is clear despite the observed high variation of individual data.
Round 2
Reviewer 2 Report
The authors addressed my questions efficiently.